# Effect of Cr on the Microstructure and Mechanical Properties of the Al-Cu-Y-Zr Alloy

Sayed M. Amer [1], Maria V. Glavatskikh [2], Ruslan Yu. Barkov [2,*], Irina S. Loginova [2] and Andrey V. Pozdniakov [2]

1   Faculty of Engineering, Mining, Metallurgy and Petroleum Engineering, Al-Azhar University, Cairo 11884, Egypt
2   Department of Physical Metallurgy of Non-Ferrous Metals, National University of Science and Technology "MISIS", 119049 Moscow, Russia
*   Correspondence: barkov@misis.ru

**Abstract:** The present investigation aimed at the determination of Cr addition on microstructure, phase composition, and mechanical properties of the Al-Cu-Y-Zr alloy. Quasi-binary alloys of the ternary Al-Cu-Y system with atomic rotation of Cu/Y = 4/1 have a narrow solidification range with high solidus temperature. The addition of 0.3% Cr in the Al-5.1Cu-1.7Y-0.3Zr alloy provides a formation of a novel quaternary $Al_{81-85}Cu_{7-10}Y_{3-4}Cr_5$ phase. $L1_2$-$Al_3(Zr,Y)$ phase spherical particles with a diameter of 50 nm were nucleated during solution treatment at 600 °C. $Al_7Cr$ precipitates were not found in the microstructure. The main strengthening effect of 32.4 MPa and 29.1 MPa was achieved from $L1_2$ and $\theta'(Al_2Cu)$ precipitates, respectively, in accordance with the calculation. The calculated hardness of 61.5HV based on the calculated $\sigma_y$ is consistent with the experimental value of hardness. $Al_3Zr$ and $Al_7Cr$ phases should be in equilibrium with (Al) in accordance with the calculated polythermal section. However, transmission electron microscopy investigation demonstrates the $Al_3(Zr,Y)$ precipitates only. As a result, the dissolved Cr atom provides a slightly higher hardness in the rolled and annealed AlCuYZrCr alloy. A suitable combination of strength and ductility was observed both after rolling and after rolling with subsequent annealing at 150 °C for 3 h—the alloy exhibited a yield strength of 308–315 MPa, an ultimate tensile strength of 323–335 MPa, and an elongation to failure of 2.0–3.3%.

**Keywords:** quasi-binary alloys; aluminum alloys; microstructure; mechanical properties; precipitates; chromium; yttrium

## 1. Introduction

Lightweight materials are becoming increasingly popular for reducing vehicle weight, saving energy, and protecting the environment. Aluminum alloys are widely used in aerospace, automobile, and other industries due to their suitable combination of mechanical, physical, and technological properties [1]. Achieving a suitable combination of operational properties is a very challenging task. Al-Si-based alloys are most used in industry to produce castings. Al-Si-based alloys have excellent casting properties, but the heat resistance and room temperature mechanical properties are low [1–5]. On the other hand, Al-Cu-based alloys have suitable mechanical properties at room temperature, the best heat resistance, and the worst casting properties in all groups of Al alloys [2–5]. For example, the solidification range in the Al-Cu-based alloys is about 100 °C, but for Al-Si-based alloys, it is less than 50 °C [4,5]. In this case, e improving f the casting properties of the Al-Cu-based alloys is a prospective aim to develop novel heat-resistant alloys.

Great interest has been shown in recent years in the investigation of the Al-Cu-based systems with a high fraction of the eutectic phases. An atypical approach to rare earth metals alloying was used by the authors of the papers [6–12]. Quasi-binary alloys of the ternary Al-Cu-X systems (X—rare earth metals (REM), such as Y, Er, Yb, Gd, and Ce) with

atomic rotation of Cu/X = 4/1 have a narrow solidification range with high solidus temperature [6–12]. A narrow solidification range provides suitable casting properties, especially a low sensitivity to hot tearing or solidification crack formation [2–5]. High solidus temperature is the first reason to make the alloys perspective for high-temperature application. The fine ((Al) + $Al_8Cu_4X$) eutectic solidify in the alloys with 4–6.5% Cu at temperatures of 600–630 °C [11–16]. $Al_8Cu_4X$ phase particles demonstrate excellent thermal stability to grow during high-temperature solution treatment [11–16]. However, the mechanical properties of ternary alloys with atomic rotation of Cu/X = 4/1 are very low [11–16]. Ternary Al-Cu-X alloys were doped by Zr, and the mechanical properties were increased due to the formation of $L1_2$-$Al_3$(Zr,X) precipitates during solution treatment [17–20]. $L1_2$-$Al_3$(Zr,X) precipitates with a size of 30–50 nm nucleated at high-temperature solution treatment of 590–605 °C [17–20]. The calculated contribution of $L1_2$-$Al_3$(Zr,X) precipitates to the yield strength of the alloy is about 27–38 MPa in the Al-Cu-Y-Zr [18], Al-Cu-Er-Zr [19], Al-Cu-Yb-Zr [20], and Al-Cu-Gd-Zr [20] alloys. Complex alloying by Zr, Mn, Mg, Ti allowed to create novel heat-resistant Al alloys [21–23]. In this case, the main strengthening effects of 54–60 MPa and 138–153 MPa in the Al-Cu-Yb- and Al-Cu-Gd-based alloys were achieved from $L1_2$ and S′ precipitates, respectively [23].

A small addition of Y (typical approach of the rare earth metals alloying) in the Al-Sc [24–26], Al-Zr [27–30], Al-Mg [31,32], Al-Cu [33], Al-Zn-Mg-Cu [34–37], and Al-Si [38–40] alloys improve the mechanical and casting properties. Yttrium substitutes a part of Sc [24–26] or Zr [27–30] in the $L1_2$-$Al_3$(Zr,X) precipitates. The best strengthening effect due to the formation of the 4-10 nm sized precipitates was achieved after annealing at about 300 °C for Al-Sc alloys and 400 °C for Al-Zr alloys. This is the main strengthening mechanism in the non-aged after quenching Al alloys [24–32]. The same strengthening mechanism is implemented in the Al-Mg alloys [31,32]. $L1_2$ precipitates formed during solution treatment may affect the strengthening during aging of the Al-Zn-Mg-Cu alloys [34–37]. Yttrium is an effective modifier of the dendritic cell in the Al-Si alloys, which increases a room temperate mechanical properties [38,39]. The formation of the fine eutectic particles with Y and Cu provides an increase in the high temperate mechanical properties of the Al-Si-based alloys [40]. Chromium is commonly used to strengthen the Al-Zn-Mg-Cu alloys due to $Al_{18}Mg_3Cr_2$ precipitates nucleation during homogenization [36,41–46]. Chromium, in combination with REM, provides additional strengthening of the Al alloys [36,41–46]. Complex alloying by Y and Cr of the Al-Cu-based alloys should be perspective to develop novel heat-resistant materials.

In this case, the aim of the present investigation is the determination of the 0.3% Cr addition on microstructure, phase composition, and mechanical properties of the Al-Cu-Y-Zr alloy in the as-cast, homogenized, and rolled conditions.

## 2. Materials and Methods

### 2.1. Alloys Preparation

The AlCuYZrCr alloy (Table 1) was melted in the resistance furnace from pure Al (99.9%) and Cu (99.5%) and Al-10Y, Al-10Cr, and Al-5Zr master alloys. The melting and pouring temperature was 750 °C. Master alloys were introduced in the Al melt separately at 750 °C. Casting was carried out into a copper water-cooling mold with a cooling rate of about 15 °C/s. The ingots dimension is $120 \times 40 \times 20$ mm$^3$. The reference AlCuYZr alloy was obtained in the same conditions.

**Table 1.** Chemical composition of the investigated alloys.

| Alloy | | Al | Cu | Y | Zr | Cr | Cu/Y |
|---|---|---|---|---|---|---|---|
| AlCuYZrCr | wt.% | bal. | 5.1 | 1.7 | 0.3 | 0.3 | - |
| | at.% | bal. | 2.27 | 0.54 | 0.09 | 0.16 | 4.2/1 |
| AlCuYZr [18] | wt.% | bal. | 4.7 | 1.6 | 0.3 | - | - |
| | at.% | bal. | 2.08 | 0.51 | 0.09 | - | 4.1/1 |

### 2.2. Microstructure and Phase Composition Analyses

The microstructure and phase composition of the alloy were investigated in detail with an optical microscope (OM) Zeiss, scanning electron microscope (SEM) TESCAN VEGA 3LMH (Tescan, Brno, Kohoutovice, Czech Republic), and transmission electron microscope (TEM) JEOL –2100 EX (Jeol Ltd., Tokyo, Japan). OM was used for grain structure investigation in the polarized light at $200\times$ and $500\times$ magnification. SEM images were obtained using in back-scattered electron (BSE) detector at $3000\times$ magnification at an operating voltage of 20 kV. SEM phase analyses were performed using by electron diffraction X-ray (EDX) detector X-max 80. Phase determination was performed using X-ray diffraction (XRD) with Cu-K$\alpha$ radiation on a Bruker D8 Advance diffractometer (Bruker, Karlsruhe, Germany).

### 2.3. Sample Preparation for Microstructure Investigation

Samples for microstructure investigation were mechanically ground and polished. Mechanical grinding and polishing were performed using Struers Labobol equipment. Grinding paper of #800, #1200, #2000, and #4000 and OP-S suspension was used for mechanical grinding and polishing. The microstructure was revealed by anodizing (15–25 V, 0–5 °C) using Barker's reagent (46 mL of HBF$_4$, 7 g of HBO$_3$, and 970 mL of H$_2$O). The average value of the grain size was measured by the random secant method in 3 images. The specimens for TEM were prepared using the A2 electrolyte on Struers Tenupol-5 equipment. The samples for TEM were mechanically ground to a 0.25 mm thickness before electrochemical preparation. XRD investigation was performed using alloy powder prepared by mechanical grinding.

### 2.4. Heat Treatment Processing

The solidus temperatures of the alloy were determined by the Labsys Setaram differential scanning calorimeter (SETARAM Instrumentation, Caluire, France) (DSC). Ingots were solution-treated at 600 °C for 1, 3, and 6 h in the Nabertherm furnace with an accuracy of about 1 °C. Aging treatment was carried out at 210 °C in the SNOL furnace with an accuracy of about 3 °C. The ingot, after solution treatment at 600 °C for 3 h and water quenching, was hot rolled at 440 °C from a thickness of 20 mm to 10 mm and at room temperature to a 1 mm thickness sheet. Rolled samples were annealed at 150, 180, 210, 250, 300, 350, 400, 450, 500, and 550 °C for different times to investigate the structure and mechanical properties evaluation.

### 2.5. Thermodynamic Calculations

Thermodynamic calculations of the multicomponent phase diagram and phase equilibria were carried out in the Thermo-Calc software in the TCAL4 database.

### 2.6. Mechanical Properties Measurements and Calculations

The hardness of the alloy samples was measured by the standard Vickers method under a 5 kg load. The hardness value HV was determined as the arithmetic mean of 5–10 measurements, and the standard deviation was calculated. The tensile samples were stretched on a Zwick/Roell Z250 Allround (Zwick/Roell, Kennesaw, GA, USA) test machine. The tensile samples with a gauge width of 6 mm and a gauge length of 22 mm were cut out from 1 mm thickness sheets. A minimum of 3 samples were tested per state, and the average value was calculated.

The hardness value of the aluminum solid solution (Al) strengthening by precipitates was also calculated using the empirical equation [4]:

$$HV = 0.18(\sigma_{ss} + \sigma_{pt} + \sigma_{Al}) + 41.7, \tag{1}$$

where $\sigma_{ss}$—contribution from solid solution; $\sigma_{pt}$—contribution from precipitates; $\sigma_{Al} = \sigma_{gb} + \sigma_d$—contribution from grain boundaries and dislocations.

In the present study, the HV was calculated, including the eutectic particles contribution: HV = $0.18\sigma_y + 41.7$, where $\sigma_y$ is the calculated yield strength of the alloy.

## 3. Results and Discussion

### 3.1. As-Cast Microstructure and Phase Composition

The Cu and Y content in the investigated AlCuYZrCr alloy is slightly higher than in the compared AlCuYZr alloy (Table 1). The main rule of the Cu/Y atomic rotation to the formation of the $Al_8Cu_4Y$ phase was observed (Table 1).

The results of the as-cast microstructure and phase composition investigation are presented in Figure 1. Primary Al solid solution dendrites (Al), fine eutectic enriched in Cu and Y, and Cu-, Y-, and Cr-rich particles identified in the as-cast microstructure (see SEM images and distribution of the alloying elements in the rectangle area in Figure 1a). The XRD pattern of the AlCuYZrCr alloy, in comparison with the XRD results of the Cr-free AlCuYZr alloy, is demonstrated in Figure 1b. (Al), $Al_8Cu_4Y$ and $(Al,Cu)_{11}Y_3$ phase peaks are clearly identified in both investigated alloys. The peaks of Cu-, Y-, and Cr-rich phase particles are not highlighted in the XRD pattern of the AlCuYZrCr alloy due to a low volume fraction or the possibility of a similar crystal structure. As a result, the peaks of Cu-, Y-, and Cr-rich phases were covered by peaks from the main phases. Point EDX SEM analyses demonstrate the content of 14.3–18.8 wt.% Cu, 9.2–9.3 wt.% Y, and 7.5–7.6 wt.% Cr (Al–bal.) in the white particles (Figure 1c,d). Recalculation of the phase composition in the atomic fraction allows us to write the phase formula as $Al_{81-85}Cu_{7-10}Y_{3-4}Cr_5$. The novel quaternary $Al_{81-85}Cu_{7-10}Y_{3-4}Cr_5$ phase was first identified in the present study.

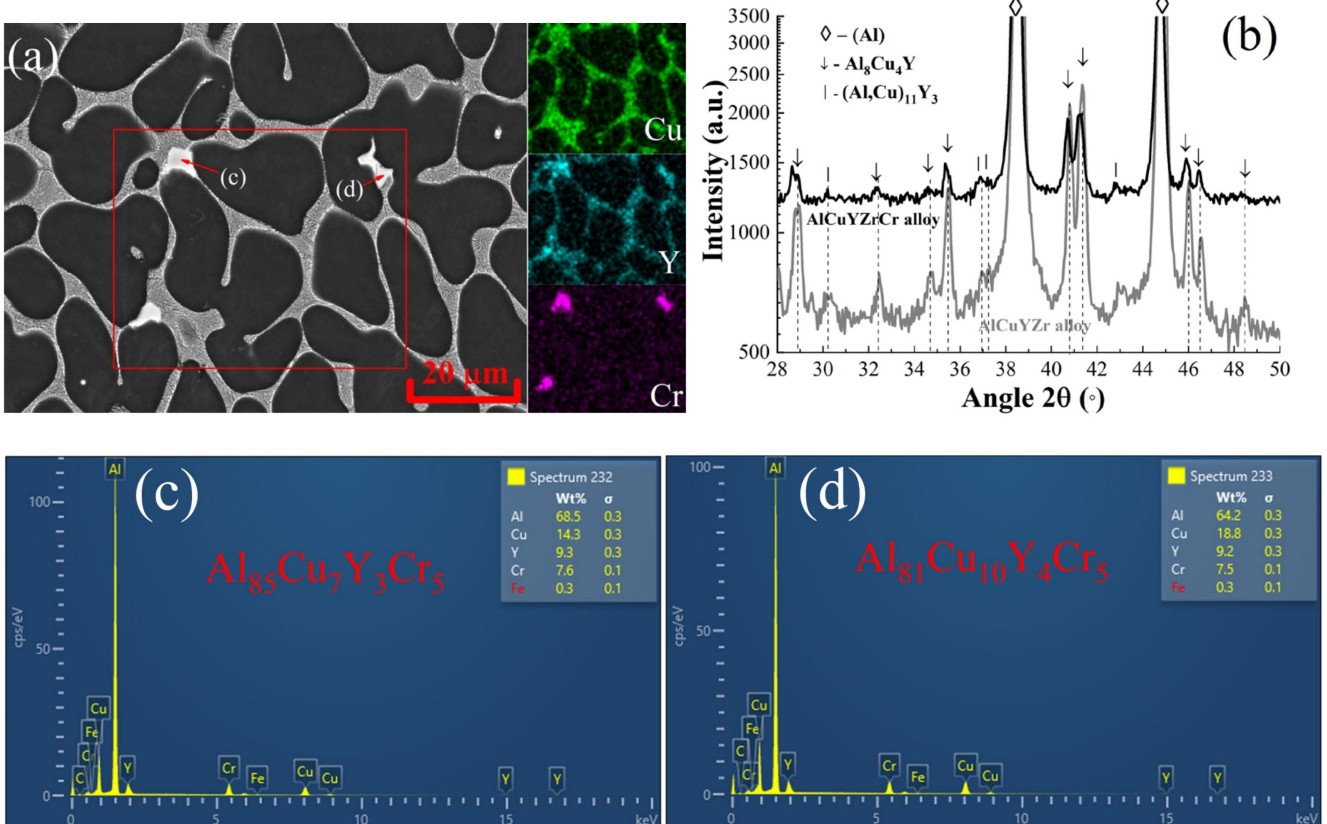

**Figure 1.** (**a**) As-cast microstructure (SEM), (**b**) XRD-patterns of the AlCuYZrCr alloy, and (**c,d**) EDX spectra from white particles (distribution of the alloying elements in the rectangle area in (**a**) and XRD pattern of the AlCuYZr alloy (gray line)).

### 3.2. Evaluation of the Microstructure under Solution Treatment

The composition of the (Al) in the as-cast and solution-treated states is presented in Table 2. The solidus temperature of the AlCuYZrCr is 611 °C in accordance with the DSC data. The solution treatment at 600 °C for 1, 3, and 6 h was applied before quenching. The content of 1.6 wt.% Cu, 0.2 wt.% Y, 0.3–0.4 wt.% Zr, and 0.3–0.4 wt.% Cr was determined in (Al) in the as-cast state by EDX SEM measurements. Two parallel processes should proceed during solution treatment in the AlCuYZrCr alloy. The first process is homogenization. The non-equilibrium part of the intermetallic phases dissolved in the (Al). As a result, the Cu content in the (Al) slightly increased to 1.8 wt.% after 3 h of solution treatment (Table 2). The equilibrium phase's particles fragmentized, spheroidized, and grew from 100 to 200 nm in the as-cast state (Figure 1a) to 200–2000 nm after 3 h of solution treatment (Figure 2). The second process is heterogenization. The supersaturated by Y, Zr, and Cr during solidification (Al) should be decomposed. The fine precipitates are clearly seen in the SEM images after 3 h of solution treatment (Figure 2). The polythermal section Al-1.8Cu-0.4Zr-(0-1)Cr was calculated to predict the phase composition of the decomposed (Al). (Al) and Al$_3$Zr phase should be in the equilibrium at 600 °C in the (Al) with a content of 0.3–0.4 wt.% Cr (Figure 3).

**Table 2.** Composition of the Al solid solution in mass.% (EDX SEM).

| State | Al | Cu | Y | Zr | Cr |
|---|---|---|---|---|---|
| As-cast | bal. | 1.6 | 0.2 | 0.3–0.4 | 0.3–0.4 |
| 600 °C for 1 h | bal. | 1.6 | 0.2 | 0.4 | 0.4 |
| 600 °C for 3 h | bal. | 1.8 | 0.2 | 0.4 | 0.4 |
| 600 °C for 6 h | bal. | 1.8 | 0.2 | 0.3 | 0.4 |

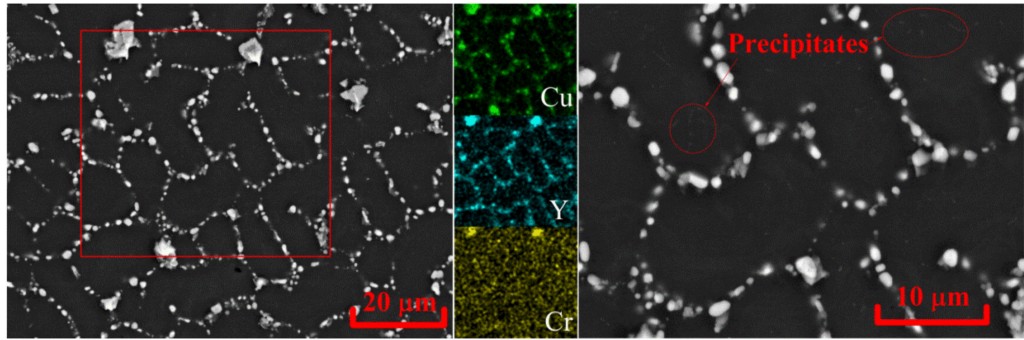

**Figure 2.** Microstructure (SEM) and distribution of the alloying elements in the rectangle area after solution treatment at 600 °C for 3 h and quenching.

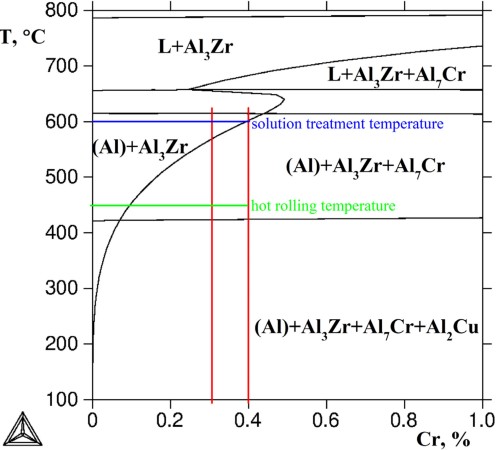

**Figure 3.** Calculated polythermal section Al-1.8Cu-0.4Zr-(0-1)Cr.

### 3.3. Phase Transformation under Solution and Aging Treatment

AlCuYZrCr alloy was solution-treated at 600 °C for 3 h, water quenched, and aged at 210 °C for 3 h. As-quenched hardness is 52 HV. The hardening effect of 4 HV was achieved after aging treatment. Very low hardening should be explained by low Cu content in the (Al). Two types of precipitates were identified in the quenched and aged conditions (Figure 4). $L1_2$-$Al_3$(Zr,Y) phase spherical particles with a diameter of 50 nm were nucleated during solution treatment at 600 °C. Typical disk-shaped θ′($Al_2$Cu) with a diameter of 120 nm and thickness of 8 nm were nucleated near $L1_2$—precipitates during aging treatment at 210 °C. θ′($Al_2$Cu) precipitates were nucleated on the $L1_2$/(Al) boundary (Figure 4c).

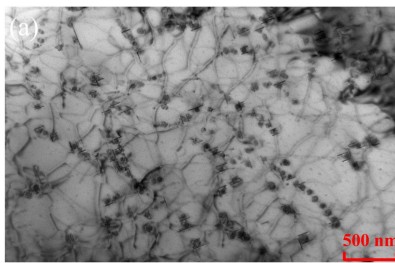
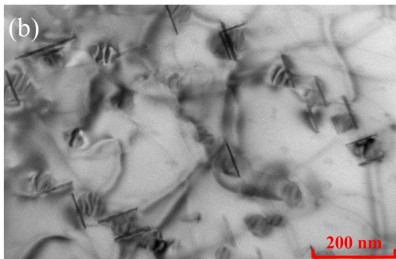
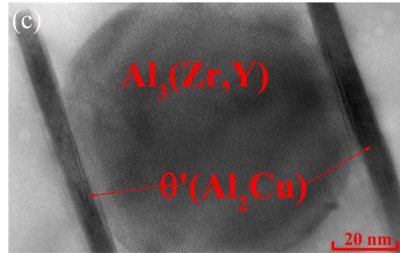

**Figure 4.** Microstructure (TEM) after solution treatment at 600 °C for 3 h, quenching and aging at 210 °C for 3 h ((**a–c**)—images at different magnification).

### 3.4. Yield Strength and Hardness Calculations

The yield strength (YS) of the investigated alloy should be predicted theoretically based on the investigated structure parameters [47–52]. In the calculation model, we will consider five strengthening mechanisms that affect the critically resolved shear stress of grains using a linear superposition:

$$\sigma_y = \Delta\sigma_{gb} + \Delta\sigma_d + \Delta\sigma_{ss} + \Delta\sigma_{ppt} + \Delta\sigma_p \tag{2}$$

where $\Delta\sigma_{gb}$—contribution from grain boundaries; $\Delta\sigma_d$—contribution from dislocations; $\Delta\sigma_{ss}$—contribution from solid solution; $\Delta\sigma_{ppt}$—contribution from precipitates; $\Delta\sigma_p$—contribution from eutectic particles.

Contribution from different structure parts was calculated by following equations:

$$\Delta\sigma_{gb} = \sigma_0 + kd^{-0.5}, \tag{3}$$

where $\sigma_0 = 10$ MPa is the friction stress, $k = 0.065$ MPa/m$^{-2}$ is the Hall–Petch slope, d is the average grain diameter;

$$\Delta\sigma_d = M\alpha_1 Gb\sqrt{\rho_{dis}}, \tag{4}$$

where M = 3 is the mean orientation factor or Taylor factor, $\alpha_1 = 0.3$ is the frequency-centered cubic lattice constant of Al, G = 26 GPa is the Shear modulus of Al, b = 0.286 nm is Burger's vector, and $\rho_{dis} = 10^9$ sm$^{-2}$ is the dislocation density;

$$\Delta\sigma_{ss} = 13.8C_{Cu}, \tag{5}$$

where $C_{Cu}$—the atomic concentration of Cu in the Al solid solution;

$$\Delta\sigma_{Orovan\_shpere} = \frac{M \cdot 0.4 \cdot Gb}{\pi\sqrt{(1-v)}} \cdot \frac{\ln\left(\frac{2\overline{R}}{r_0}\right)}{\lambda}, \tag{6}$$

where $r_0 = 1.5b$, $v = 0.345$ is the Poisson's ratio, $\overline{R} = \frac{\pi R_S}{4}$ is the mean planar radius (where Rs is the precipitate radius), $\lambda = R_S \cdot \left(\sqrt{\frac{2\pi}{3f}} - \frac{\pi}{4}\right)$ is the edge-to-edge interprecipitate spacing (where $f$ is the volume fraction of the precipitates);

$$\Delta\sigma_{Orovan\_disc} = 0.13\left(\frac{Gb}{\sqrt{d \cdot w}}\right)\left(\sqrt{f} + 0.75 \cdot \sqrt{\frac{d}{w}} \cdot f + 0.14 \cdot \frac{d}{w} \cdot f^{3/2}\right) \cdot \left(\ln\left(\frac{0.87 \cdot \sqrt{d \cdot w}}{r_0}\right)\right) \tag{7}$$

where d—precipitate diameter, w—precipitate thickness, f—volume fraction of the precipitates.

The calculated contribution from different structure parts is summarized in Table 3. The volume fractions of the $L1_2$ and $\theta'$ precipitates were calculated from Al-Zr, Al-Y, and Al-Cu-Zr-Cr phase diagrams. Zr and Y content in the Al solid solution are about 0.4 and 0.2 wt% respectively (Table 2). The volume fraction of the complex $L1_2$-$Al_3$(Zr,Y) phases can be estimated by considering the density of the simple $L1_2$-$Al_3Zr$ and $L1_2$-$Al_3Y$ phases, which density is 4.17 [53] and 3.66 g/cm$^3$ [54] respectively. The general volume fraction of the $L1_2$-$Al_3$(Zr,Y) precipitates is 0.005. The size of grains, eutectic particles, and precipitates was calculated from OM, SEM, and TEM images. The average grain size of the alloy is 500 µm (Figure 5). A dislocation density of $10^9$ sm$^{-2}$ [4] is typical for homogenized and quenched alloys. The average diameter of 1 µm and volume fraction of 0.08 of the eutectic particles was measured from SEM images by the secant method. The volume fraction of $\theta'$($Al_2Cu$) was calculated from the phase diagram as a fraction of $\theta$ phase at an aging temperature in the Al-Cu-Zr-Cr system.

**Table 3.** Calculated contribution from different structure parts.

| Equation | Structure Parameters | Contribution, MPa |
|---|---|---|
| $\Delta\sigma_{gb} = \sigma_0 + kd^{-0.5}$ [47–50] | $\sigma_0 = 10$ MPa, k = 0.065 MPa/m$^{-2}$, d = 500 ± 80 µm | 12.9 |
| $\Delta\sigma_d = M\alpha_1 Gb\sqrt{\rho_{dis}}$ [47] | $\rho_{dis} = 10^9$ sm$^{-2}$ [4] | 21 |
| $\Delta\sigma_{ss} = 13.8C_{Cu}$ [49] | $C_{Cu} = 0.1\%$ | 2.5 |
| $\Delta\sigma_p$(Orovan equation [49]) | Eutectic particles (r = 500 nm, f = 0.08) | 12.3 |
| $\Delta\sigma_{ppt}$ (Orovan equations for disc-shaped particles [52]) | $L1_2$ (r = 25 nm, f = 0.005) | 32.4 |
| | $\theta'$($Al_2Cu$) (d = 120 nm, h = 6 nm, f = 0.02 | 29.1 |
| | $\sigma_y$ | 110 |
| $HV_{calc} = 0.18 \cdot \sigma_y + 41.7$ [4] | | 61.5 |
| | $HV_{exp}$ | 56 |

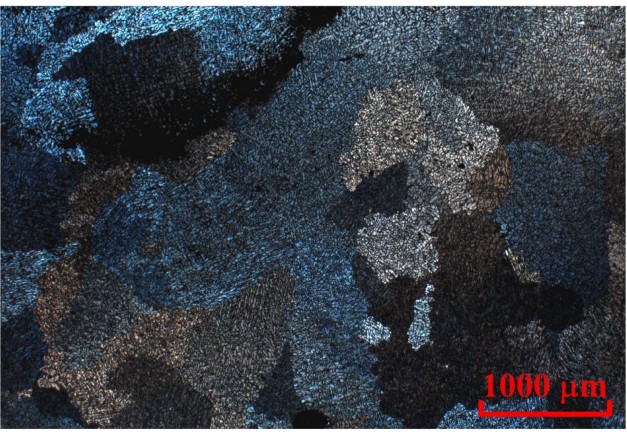

**Figure 5.** The grain structure of the AlCuYZrCr alloy in the as-cast state (OM).

The main strengthening effect of 32.4 MPa and 29.1 MPa was achieved from L1$_2$ and θ′(Al$_2$Cu) precipitates, respectively. The calculated σ$_y$ value is 110 MPa. The calculated hardness value of 61.5 HV based on the calculated σ$_y$ is consistent with the experimental value of alloy hardness (Table 3). The accuracy of the calculated HV is 9.8%.

### 3.5. Microstructure and Hardness Evaluation after Rolling and Annealing

Evaluation of the HV depends on the time and temperature of annealing for 1 h of the AlCuYZrCr rolled alloy in comparison with AlCuYZr alloy is presented in Figure 6. AlCuYZrCr alloy has a 104 HV in the as-rolled state and demonstrates a slightly lower softening during annealing than AlCuYZr alloy (Figure 6). Two processes may take place in the Al-Cu-based alloy annealed at low temperatures after hot and cold rolling. The first process is softening due to the recovery and polygonization process. The second process is aging due to the possibility of the supersaturated solid solution formation after hot rolling. The aging hardening in the investigated alloy is very low, as was demonstrated earlier. AlCuYZrCr alloy demonstrates the same tendency to the softening but has a slightly higher hardness in the annealed after-rolling state. Annealing of the rolled sheet at 150–210 °C for 2–5 h provides a hardness of 92–98 HV (Figure 6a). At the same time, the hardness of the Cr-free alloy is 87–95 HV after annealing at 150–180 °C. Increasing the annealing temperature to 300 °C saves the non-recrystallized structure in both alloys with the same tendency of softening (Figure 6b). Subgrain structure with low angular boundaries formed in the alloy AlCuYZrCr alloy in the annealed at 300 °C for 1 h (Figure 7). Sub-grains with a size of 200–500 nm were formed. The L1$_2$-Al$_3$(Zr,Y) is an effective restrain dislocation sub-boundaries (right image in Figure 7).

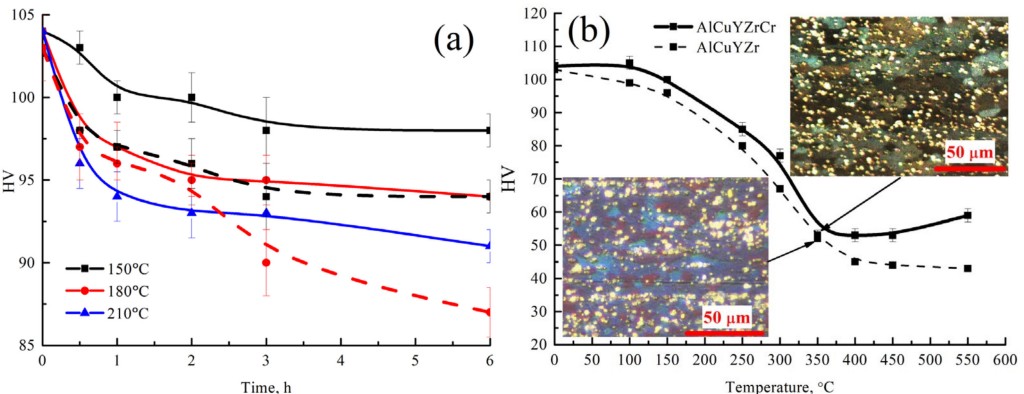

**Figure 6.** Evaluation of the HV depends on the (**a**) time and (**b**) temperature of annealing for 1 h of the AlCuYZrCr alloy in comparison with the AlCuYZr (dash lines) alloy.

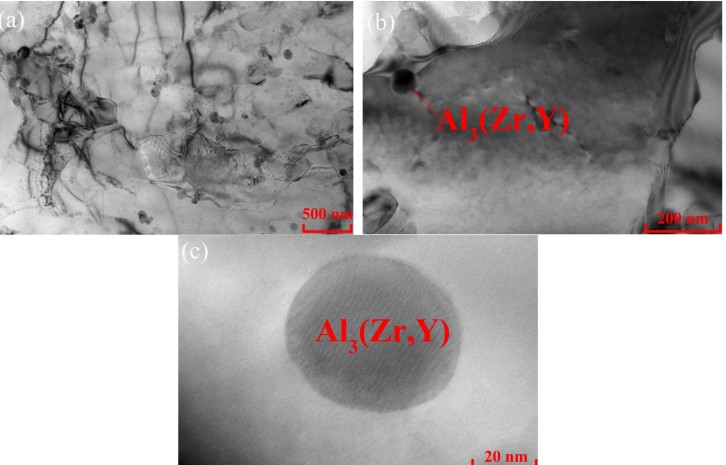

**Figure 7.** Microstructure (TEM) after solution treatment at 600 °C for 3 h, hot rolling at 440 °C, cold rolling, and annealing at 300 °C for 1 h ((**a–c**)—images at different magnification).

Recrystallized grains were found in OM after 1 h of annealing at 350 °C (inserts in Figure 6b). According to our assumption, the $Al_7Cr$ precipitates should be formed during hot rolling at 440 °C. Assumption based on the thermodynamic calculation (Figure 3). $Al_3Zr$ and $Al_7Cr$ phases should be in equilibrium with (Al) in accordance with the calculated poly-thermal section (Figure 3). Formation of the $Al_7Cr$ precipitates should provide additional strengthening and increase the recrystallization temperature. However, the TEM investigation demonstrates the $Al_3(Zr,Y)$ precipitates only (Figure 7). For example, the addition of Mn provides a significant strengthening and increases the recrystallization temperature of the AlCuYZr alloy [55]. As a result, dissolved Cr atoms and $Al_{81-85}Cu_{7-10}Y_{3-4}Cr_5$ phase particles of the solidification origin provide a slightly higher hardness in the rolled and annealed AlCuYZrCr alloy. Investigation of the Cr content and solution treatment modes is necessary to provide the formation of the $Al_7Cr$ precipitates and strengthening. In addition, it should be indicated a slightly increasing in hardness with increasing annealing temperature from 400 to 550 °C. This phenomenon may have two explanations. The first explanation is based on the natural aging process, which may take place after fast cooling of the 1 mm thickness sheets from high temperatures. The second explanation is based on the possibility of the decomposition of the Al solid solution with the formation of additional $L1_2$-$Al_3(Zr,Y)$ precipitates. High dislocation density after cold rolling may accelerate the precipitate's nucleation because both homogeneous and heterogeneous nucleation mechanisms were identified for $L1_2$-$Al_3(Zr,Y)$ precipitates [18]. The same effect was indicated in the Al-Cu-Y-Zr alloy with Mn addition [55].

Table 4 illustrates the results of the tensile tests of the as-rolled and annealed alloy samples. Higher strength values were achieved in the AlCuYZrCr alloy. A suitable combination of strength and ductility was observed both after rolling and after rolling with subsequent annealing at 150 °C for 3 h—the alloy exhibited a yield strength of 308–315 MPa, an ultimate tensile strength of 323–335 MPa, and an elongation to failure of 2.0–3.3%. A lower strength was achieved in the AlCuYZr alloy in the same condition: yield strength of 270–303 MPa and ultimate tensile strength of 299–328 MPa [18]. For comparison, the sheets of the wrought Al alloy D16 (Al-Cu-Mg-Mn-based system alloy) in the rolled and annealed states have a yield strength of 230–360 MPa and ultimate tensile strength of 365–475 MPa and an elongation to failure of 8–13%. Complex alloying by Zr, Cr, Mg, and Ti should provide the achievement of significantly higher mechanical properties and create novel Al alloys.

**Table 4.** Tensile properties of the AlCuYZrCr and AlCuYZr [18] in the as-rolled and annealed states.

| State | YS, MPa | UTS, MPa | El., % |
|---|---|---|---|
| AlCuYZrCr | | | |
| As rolled | $315 \pm 5$ | $335 \pm 9$ | $2 \pm 1$ |
| Annealed at 150 °C for 3 h | $308 \pm 2$ | $323 \pm 3$ | $3.3 \pm 0.6$ |
| Annealed at 180 °C for 3 h | $302 \pm 1$ | $312 \pm 6$ | $0.9 \pm 0.3$ |
| Annealed at 210 °C for 2 h | $265 \pm 7$ | $292 \pm 5$ | $1.2 \pm 0.2$ |
| AlCuYZr [18] | | | |
| As rolled | $303 \pm 2$ | $328 \pm 3$ | $5.0 \pm 0.3$ |
| Annealed at 100 °C for 1 h | $292 \pm 3$ | $320 \pm 5$ | $5.3 \pm 0.2$ |
| Annealed at 100 °C for 3 h | $287 \pm 1$ | $317 \pm 3$ | $4.8 \pm 0.6$ |
| Annealed at 150 °C for 1 h | $270 \pm 4$ | $302 \pm 1$ | $4.8 \pm 0.7$ |
| Annealed at 150 °C for 3 h | $270 \pm 3$ | $299 \pm 3$ | $6.3 \pm 0.3$ |

## 4. Conclusions

Microstructure, phase composition, and mechanical properties of the Al-5.1Cu-1.7Y-0.3Zr alloy with 0.3% Cr addition were investigated by optical, scanning, and transmission electron microscopy, hardness measurements, and calculations. The addition of Cr in the Al-Cu-Y-Zr alloy provides the formation of a novel quaternary $Al_{81-85}Cu_{7-10}Y_{3-4}Cr_5$ phase. $L1_2$-$Al_3(Zr,Y)$ phase spherical particles with a diameter of 50 nm were nucleated during

solution treatment at 600 °C. Al$_7$Cr precipitates were not found in the microstructure. The main strengthening effect of 32.4 MPa and 29.1 MPa was achieved in the quenched and aged alloy from L1$_2$ and θ′(Al$_2$Cu) precipitates, respectively, in accordance with the calculation. The calculated hardness of 61.5HV based on the calculated σ$_y$ is consistent with the experimental value of hardness after aging treatment. Al$_3$Zr and Al$_7$Cr phases should be in equilibrium with (Al) in accordance with the calculated polythermal section. However, transmission electron microscopy investigation demonstrates the Al$_3$(Zr,Y) precipitates only. As a result, dissolved Cr atoms provide a slightly higher hardness in the rolled and annealed AlCuYZrCr alloy. Investigation of the Cr content and solution treatment modes is necessary to provide the formation of the Al$_7$Cr precipitates and strengthening. A suitable combination of strength and ductility was observed both after rolling and after rolling with subsequent annealing at 150 °C for 3 h—the alloy exhibited a yield strength of 308–315 MPa, an ultimate tensile strength of 323–335 MPa, and an elongation to failure of 2.0–3.3%.

**Author Contributions:** Conceptualization, A.V.P.; methodology S.M.A.; formal analysis, S.M.A. and A.V.P.; investigation, M.V.G. and A.V.P.; data curation, S.M.A., R.Y.B., and I.S.L.; writing—original draft preparation, A.V.P.; writing—review and editing, A.V.P.; visualization, M.V.G. and A.V.P.; supervision, R.Y.B. and A.V.P.; funding acquisition, R.Y.B. All authors have read and agreed to the published version of the manuscript.

**Funding:** The work was supported by the Russian Science Foundation (project no. 19-79-10242), https://rscf.ru/project/19-79-10242/.

**Data Availability Statement:** Not Applicable.

**Conflicts of Interest:** The authors declare no conflict of interest.

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
