# Peer review of "Effect of Cr on the Microstructure and Mechanical Properties of the Al-Cu-Y-Zr Alloy"

_metals, doi:10.3390/met13020349_

Round 1

Reviewer 1 Report

This paper reports effect of Cr on the microstructure and mechanical properties of 2 the Al-Cu-Y-Zr alloy. Some comments are listed below:

1.       In abstract (line 19), “As-a result,” should be “As a result,”

2.       In line 99-100, “The grain structure of as-cast samples was investigated using OM under polarized light.” should be deleted.

3.       In line 153, 271, “As a results The” should be “As a result, the”

4.       Fig. 4 is not mentioned in the main text. What is the difference of Fig. 4a,b,c. In addition, Fig. 4c cannot support the nucleation of Al2Cu on Al3Zr. They may just connect together.

5.       The calculation of volume fraction is not convincing. More details are needed.  

6.       “The average grain size of the alloy 234 is 500 μm (image is not present in the manuscript).” It is necessary to include grain size here.

7.       In line 257, “id 87-95 HV” should be “is 87-95 HV”  

8.       In line 258, “save” should be “saves”

9.       In line 271, 318, “As-a result” should be “As a result”  

10.   In line 274, “shuold be indicated” should be “It should be indicated”

11.   In line 448, “54.” should be deleted. In line 450, “55.” should be deleted.

Author Response

Thanks for the detailed review of our article!
a detailed response to your comments is presented in the attached file.

Reviewer 2 Report

1. In the introduction, the application status of Cr element in Al alloy should be emphasized to find out the existing problems.

2. In Al-Cu-Y-Zr, how does the form of Cr element evolve in the process of as-cast, solid solution, rolling and post-rolling annealing?What is the influence mechanism of Cr element on alloy properties?

3. The author claims that the Al81-85Cu7-10Y3-4Cr5 quaternary phase has been identified for the first time, but the research is not sufficient. It is necessary to determine its crystal structure by various means, such as transmission electron microscope, high resolution, XRD, etc., and conduct cross validation.

4. Table 2 has no units, and the units should be supplemented.

5. The microstructure and properties after rolling shall be supplemented.

6. It is said in the manuscript that Cr improves the strength and toughness of the alloy. However, it can be seen from Table 4 that Cr improves the strength of the alloy, but significantly reduces the plasticity. The author said that the alloy with Cr has good strength and toughness compatibility, which is worth discussing.

Author Response

Thanks for the detailed review of our article!
А detailed response on your comments is presented in the attached file.

Reviewer 3 Report

The microstructure and mechanical properties of a quinary aluminium alloy are described. The work is rigorous and sufficiently detailed. Experimental observations are linked to theoretical and empirical calculations. The paper is worth publishing in Metals.

In the Introduction, it is stated that the aimed application of the alloy is for casting of strong and heat-resistant materials. To link the results with the claimed purposes of the study, it would help to discuss them in view of industrial interest. Is the new alloy expected to be more heat resistant? Does it have properties necessary for the casting process? Is the increase in the ultimate tensile strength compared to the quaternary reference alloy shown in Table 4 sufficient for adopting the material in technical practice?

The authors claim that they found a new phase, Al81-85Cu7-10Y3-4Cr5. It is based on EDX spot analyses. This cannot be considered as a proof of the existence of the phase. Due to the penettration depth, EDX can easily „see“ elements under the analysed region. It can also be a fine-grained mixture of several phases. A multianalytical approach including XRD would need to be applied to clearly prove it is a phase. The formulation should be changed to reflect the current uncertainty.

The English needs to be improved. Although the text is basically understandable, there are sentences with unclear meaning and many typos. I recommend the authors hiring a professional proofreader, who will help them to improve the manuscript and make it easier to read and follow for readers. Several examples follow:

-       Line 19. As-a result.

-       Line 53. …alloys was alloyed…

-       Line 62. In addition small additive…

-       Line 66. …effect due formation…

-       Line 68–69, 146-148, 265. Verb is missing.

-       Line 82. …alloy with compositions…

-       Line 96. …scanned in back-scattered electron (BSE) detector…

-       Line 97 and elsewhere. …performed using by…

-       Line 120-121. …investigate the structure and mechanical properties evaluation.

-       Line 138. …calculated with including…

-       Line 166. …in accordance the…

-       Line 173. …sheroidized…

-       Line 202. …be predicated…

-       Line 251. …poliginization process.

-       Line 257. …Cr free alloy id…

-       Line 259. …the tendence of softening…

-       Line 264 and elsewhere. …precipitaties shuold…

-       Line 271. …dissolved Cr atom provide…

-       Line 273. …treatment regims…

-       Line 274. …a sligntly increasing…

-       Line 275. This phenomena…

-       Line 296. …alloy was achieved a lower strength…

-       Line 314. …in accordance the calculation.

-       Line 316. …in accordance the calculated…

Author Response

(The authors gave the same response as above.)

Round 2

Reviewer 2 Report

1. Since the title is "Effect of Cr on the microstructure and mechanical properties of the Al-Cu-Y-Zr alloy", the full text should focus on the effects of Cr on the microstructure and properties of the alloy. According to the data in Table 4, the strength of AlCuYZr added with Cr increased but the plasticity decreased significantly, so the conclusion of "A good combination of strength and ductility was observed" (On line 299) in the text could not be obtained. The data in Tab.4 can not support the conclusion, which has nothing to do with the fact that the content of the main elements Cu and Y in the investigated alloy is higher than in the reference alloy.

2.  The predicted value of yield strength in Section 3.4 is 110MPa, which is greatly different from the measured value (about 300MPa) in Table 4.

Author Response

Dear Reviewer, thank you for a detailed analyse of our investigation and presented comments.

  1. Since the title is "Effect of Cr on the microstructure and mechanical properties of the Al-Cu-Y-Zr alloy", the full text should focus on the effects of Cr on the microstructure and properties of the alloy. According to the data in Table 4, the strength of AlCuYZr added with Cr increased but the plasticity decreased significantly, so the conclusion of "A good combination of strength and ductility was observed" (On line 299) in the text could not be obtained. The data in Tab.4 can not support the conclusion, which has nothing to do with the fact that the content of the main elements Cu and Y in the investigated alloy is higher than in the reference alloy.

Throughout the article we demonstrate the comparison the structure and properties of the Cr doped alloy with Cr-free alloy. The higher content of Cu and Y provide a higher volume fraction of the brittleness intermetallic phases of the solidification origin. This is the first possible reason to decreasing the plasticity.

We just divide the highest strength and ductility in the rolled and annealed state when wrote “A good combination of strength and ductility”. The plasticity is not high, not the best or excellent. The plasticity is low but in the combination with high strength we say about good properties.

  1. The predicted value of yield strength in Section 3.4 is 110MPa, which is greatly different from the measured value (about 300MPa) in Table 4.

The calculated value of the YS was predicted for quenched and aged condition of the alloys ingot. The YS of 300 MPa was obtained in the cold rolled state in the sheet.